# Tumor Budding T-cell Graphs: Assessing the Need for Resection in pT1 Colorectal Cancer Patients

**Linda Studer**[1,2,3]                                    LINDA.STUDER@UNIFR.CH
[1] *Document, Image and Video Analysis Research Group, University of Fribourg, Switzerland*
[2] *iCoSyS, University of Applied Sciences and Arts Western Switzerland, Fribourg, Switzerland*
[3] *Institute of Tissue Medicine and Pathology, University of Bern, Switzerland*

**John-Melle Bokhorst**[4]                            JOHN-MELLE.BOKHORST@RADBOUDUMC.NL
[4] *Department of Pathology, Radboudumc, Nijmegen, Netherlands*

**Iris Nagtegaal**[4]                                    IRIS.NAGTEGAAL@RADBOUDUMC.NL
**Inti Zlobec**[3]                                          INTI.ZLOBEC@UNIBE.CH
**Heather Dawson**[3]                                  HEATHER.DAWSON@UNIBE.CH
**Andreas Fischer**[1,2]                              ANDREAS.FISCHER@HEFR.CH

**Editors:** Accepted for publication at MIDL 2023

## Abstract

Colon resection is often the treatment of choice for colorectal cancer (CRC) patients. However, especially for minimally invasive cancer, such as pT1, simply removing the polyps may be enough to stop cancer progression. Different histopathological risk factors such as tumor grade and invasion depth currently found the basis for the need for colon resection in pT1 CRC patients. Here, we investigate two additional risk factors, tumor budding and lymphocyte infiltration at the invasive front, which are known to be clinically relevant. We capture the spatial layout of tumor buds and T-cells and use graph-based deep learning to investigate them as potential risk predictors. Our pT1 Hotspot Tumor Budding T-cell Graph (pT1-HBTG) dataset consists of 626 tumor budding hotspots from 575 patients. We propose and compare three different graph structures, as well as combinations of the node labels. The best-performing Graph Neural Network architecture is able to increase specificity by 20% compared to the currently recommended risk stratification based on histopathological risk factors, without losing any sensitivity. We believe that using a graph-based analysis can help to assist pathologists in making risk assessments for pT1 CRC patients, and thus decrease the number of patients undergoing potentially unnecessary surgery. Both the code and dataset are made publicly available.

**Keywords:** Digital Pathology, Graph Neural Networks, Colorectal Cancer

## 1. Introduction

Thanks to screening programs becoming more common, colorectal cancer (CRC) is getting diagnosed in earlier, and more easily treatable stages. pT1 is such a form of early CRC, only invading the submucosa, but not yet the muscle layer. Thus, defining appropriate treatment guidelines is of high importance. Even though these cancers are minimally invasive, they can still cause lymph node metastasis, which puts patients at risk for distant metastasis (Burt et al., 2010; Argilés et al., 2020). That, however, cannot be assessed just from the polypectomy, but only after surgical colon resection (also referred to as a colectomy).

Among pT1 CRC patients, less than 15% actually have metastasis present in lymph nodes at the time of diagnosis (Nivatvongs et al., 1991; Kobayashi et al., 2012; Dykstra et al., 2021). Cancer registry data[1] in Bern, Switzerland from 2014 show that 55% of all pT1 CRC patients nonetheless underwent a colectomy. (Zwager et al., 2022) find an even higher disparity, from the 70%–80% patients considered as high risk, more than 90% are lymph node negative. This is due to current clinical guidelines (Bosch et al., 2013; Argilés et al., 2020) commonly recommending that patients should undergo surgery in the presence of just one unfavorable histological risk factor. Hence, improving this risk assessment has a high potential to reduce overtreatment and potential clinical complications (Vermeer et al., 2019; Zwager et al., 2022).

Deep learning has provided solutions to a variety of challenges in pathology in recent years (Echle et al., 2021; Gurcan et al., 2009), such as tissue segmentation (Abbet et al., 2022), survival analysis (Bychkov et al., 2018; Abbet et al., 2020), and predicting lymph node metastasis (Kiehl et al., 2021; Brockmoeller et al., 2022; Khan et al., 2023). Lately, especially the combination of deep learning with spatial analysis has become of high interest (Jaume et al., 2021; Ahmedt-Aristizabal et al., 2021; Pati et al., 2022). Graph-based representations allow us to capture the spatial layout of the tissue and cells of interest and map their interactions in a very concise way. Thus, they make it possible to focus the deep learning algorithm on entities with known histopathological and clinical importance.

Two such factors that are known to play an important role in CRC are tumor budding and CD8+ lymphocytes (also known as cytotoxic T-cells), especially at the invasive front (Lugli et al., 2009). Tumor budding is a well-established prognostic factor in CRC (Lugli et al., 2017; Studer et al., 2021a). Several studies have also highlighted the prognostic power of T-cell scoring (Alwers et al., 2022). Most studies actually investigate the T-cell scoring in relation to tumor budding (Dawson et al., 2020; Nearchou et al., 2021; Lugli et al., 2009; Nearchou et al., 2019; Studer et al., 2022). However, so far the analysis has been limited to creating statistical scores, based on the ratio of counts in areas or within a radius, and other spatial statistics, and are performed on multi-stage cohorts.

To our knowledge, this is the first work exploring the interaction between tumor buds and T-cells using a graph-based approach. Another contribution is our investigation of different graph representations and making the dataset publicly available. Additionally, we aim to address a question of high clinical importance, as risk assessment of polypectomy specimens will only become more important in the future.

The paper is structured as follows: Section 2 describes the used Graph Neural Networks (GNNs) methods, Section 3 introduces the dataset, and Section 4 provides details on the experimental setup and presents the results. Finally, Section 5 summarizes our findings.

## 2. Graph Neural Networks

With the introduction of GNNs, the field of deep learning has been expanded into the domain of graph-based analysis. Mathematically, a graph $G$ is defined as a tuple of $(N, E, \alpha, \beta)$, where $N$ denotes a finite set of nodes (or vertices), and $E$ a set of edges, which are inserted according to a specified edge insertion function, and $\alpha$ and $\beta$ are the node and edge labeling functions, respectively.

---

1. https://www.krebsregister.unibe.ch

For classification tasks, GNNs architectures can be divided into two phases, a message-passing (MP) and a read-out phase. During the MP phase, information is exchanged between connected nodes, and then aggregated to update their hidden state:

$$\mathbf{a}_i^{k+1} = AGGREGATE_{j \in \mathcal{N}(i)}(\mathbf{x}_i^k, \mathbf{x}_j^k, \mathbf{e}_{ij}^k) \tag{1}$$

$$\mathbf{x}_i^{k+1} = UPDATE(\mathbf{x}_i^k, \mathbf{a}_i^{k+1}), \tag{2}$$

where $\mathcal{N}(i)$ denotes the neighbouring nodes of node $i$, $x_i$ the feature vector of node $i$, $e_{ij}$ the edge feature vector between node $i$ and $j$. In the first round (or layer) of MP, the hidden node state is the node feature vector of the input graph. By performing $k$ rounds (or layers) of MP, the aggregation range is extended to the $k$-hop neighborhood.

Over the years, many different MP functions have been introduced, however, the performance of a GNN model is highly dependent on the graph type and the right choice is not trivial (Thomas et al., 2022; You et al., 2020). We use and compare three of the most commonly used ones for graph classification, namely GraphSAGE, GIN, and GATv2. They are all based on graph convolution, more details are provided in Appendix B.

In order to perform graph classification, the information of all hidden node states is aggregated and passed to the classification header, which is often a multi-layer perceptron (MLP). This is referred to as the read-out phase:

$$\mathbf{v}_G = READOUT(\mathbf{x}_i^K | i \in G). \tag{3}$$

where $\mathbf{x}_i^K$ is the node state after the last MP round. However, it has been shown to be beneficial to also consider the intermediate representations, i.e. performing a read-out after each layer. These skip connections were introduced by (Xu et al., 2018b) and termed Jumping Knowledge (JK). The different read-out vectors can e.g. be concatenated such as $\mathbf{v}_G = \mathbf{v}^1 \parallel \ldots \parallel \mathbf{v}^K$. The first part of Figure 1 gives an overview of the GNN architecture.

## 3. pT1 Hotspot Tumor Budding T-cell Graph (pT1-HBTG) Dataset

The graph creation workflow is depicted in Figure 1. Figure 2 shows an example image for each graph representation and class. The pT1-HBTG dataset is available open-source[2], additional details on the dataset and more example images can be found in Appendix C.

### 3.1. pT1 Patient Cohort

Our pT1 cohort consists of patients that, after the polypectomy, either (A) underwent resection, and thus have a known lymph node status, or (B) have at least 36 months of follow-up information about local/distant recurrence. We categorize the patients as either high- or low-risk, according to the clinical question of whether, in retrospect, they required the colectomy. Patients (A) without lymph node metastasis (N0) and patients (B) without recurrence are considered low-risk. Thus, patients (A) with lymph node metastasis (N+) and patients (B) with a recurrence are grouped into the high-risk class.

In total, we have 626 whole slide images (WSIs) of polyps from 575 pT1 CRC patients collected from eight different pathology institutes and selected by expert pathologists. The

---

2. https://zenodo.org/record/7867085

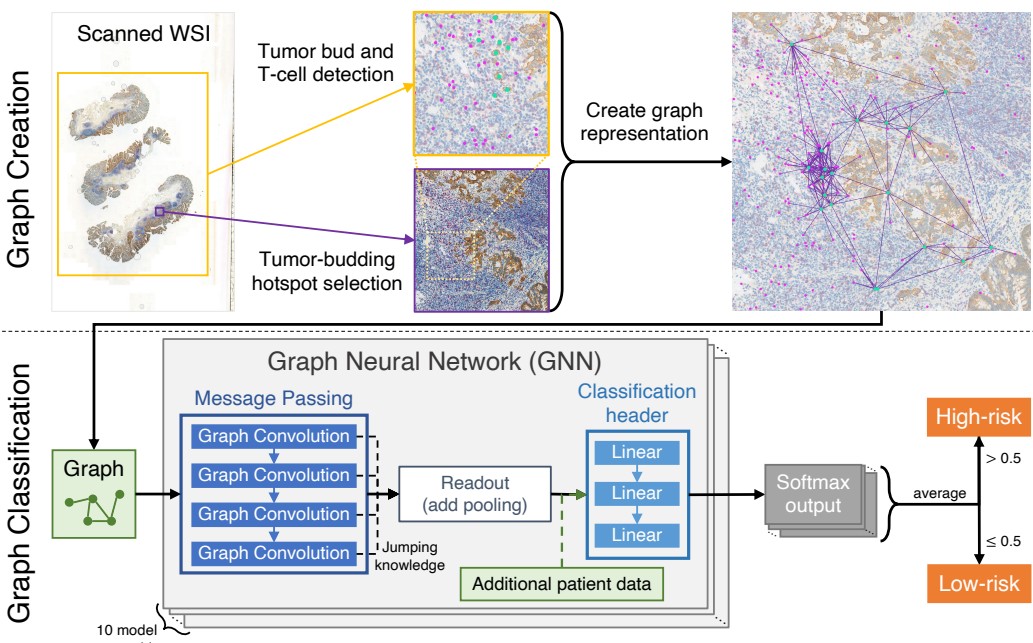

Figure 1: Schematic overview of the framework. To build the graph, tumor buds and T-cells are automatically detected using CNNs, and the tumor budding hotspot is selected by a pathologist. All detected objects within it are then used as nodes to build the graph representation (tumor bud detections are highlighted in green, and T-cells in pink). The example shown here is the *Delaunay-Star* edge configuration. The graph classification GNN consists of a message passing phase, followed by a readout phase, and a classification MLP. An ensemble of 10 models is used to make the risk classification, based on the averaged softmax output.

WSIs are double-stained for CD8-AE1/AE3 using immunohistochemistry (AE1-AE3 pancytokeratin for the tumor, and CD8 for the cytotoxic T-cells). They are digitized using a Pannoramic 250 scanner at $0.243\mu m/pixel$. The class distribution is unbalanced, with 541 versus 85 for the low-risk and high-risk groups, respectively.

### 3.2. Graph Representations

The pT1-HBTGs represent the tumor budding hotspot of a slide according to the IT-BCC (Lugli et al., 2017) recommendations (area of highest budding at the invasive front, $0.785mm^2$), selected by an expert pathologist. As introduced in section 2, a graph consists of *nodes* and *edges*, which are inserted according to a specified edge insertion function. Both nodes and edges can have additional information attached to them, also referred to as *labels*. The pipeline for graph creation is available on GitHub[3].

---

3. https://github.com/digitalpathologybern/BT-graph-creation

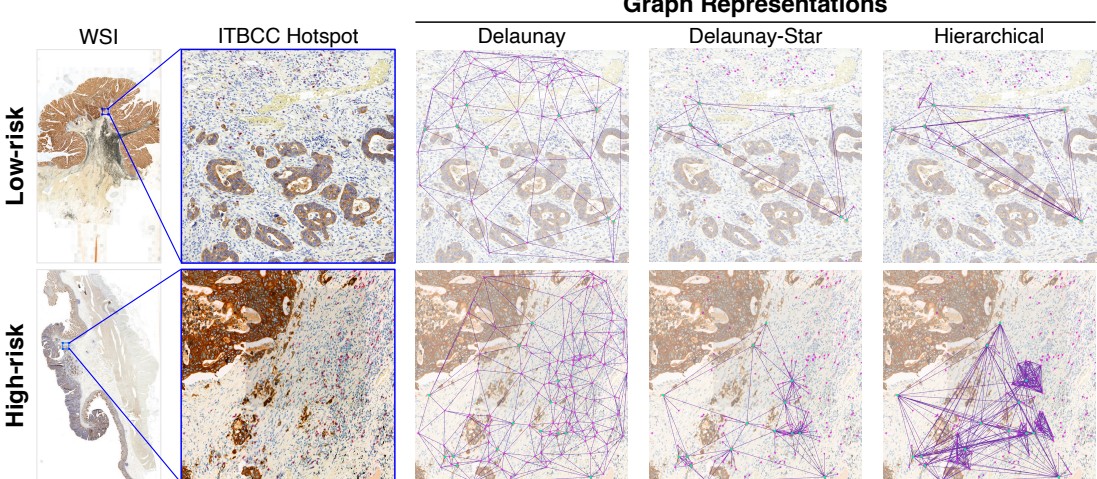

Figure 2: Example graph representations for a low- and a high-risk patient. The nodes are color-coded based on the *type* node label, tumor buds in green and T-cells in pink. The blue rectangle on the WSI shows the annotation of the ITBCC hotspot.

### 3.2.1. Node Detection and Node Labeling

Since our graphs map the interaction of tumor buds with T-cells, they build the nodes of our graphs. T-cells are single cells, whereas a tumor bud is defined as a cluster of 1-4 tumor cells. The tumor bud detection is performed using a student-teacher CNN (Bokhorst et al., 2022) model, which is available on Grand Challenge[4]. The algorithm used for the T-cells is based on U-Net, with additional post-processing (Gaussian filtering and regional maxima detection) (Swiderska-Chadaj et al., 2019). Color deconvolution is performed as a pre-processing step as the algorithm is trained on single-stain immunohistochemistry (IHC). The quality of the output was manually reviewed by an expert pathologist on a subset, with a precision of 89.9% and recall of 93.0% (Studer et al., 2020).

We define three different types of node features. First, the *type* of the node, i.e. tumor bud or T-cell, which is one-hot encoded. Second, we use the *coordinates* of the node, normalized to the center of the hotspot using zero centering: $x' = x - \bar{x}$. Third, we compute the *embedding vector* of crops ($200 \times 200$ *pixels* centered on the node coordinates) using the ImageNet (Deng et al., 2009) pre-trained DINO model (Caron et al., 2021). Using the ViT-16 backbone, we get a feature vector of size 384.

### 3.2.2. Edge Insertion and Edge Labelling

As an edge label, we use the distance between the nodes in $\mu m$. We define three different edge insertion functions. As a baseline, we use the *Delaunay* triangulation (Delaunay and Spherevide, 1934) (without distinguishing between the two node types), which is very commonly used in histopathology, especially for cell graphs (Sharma et al., 2015). We only consider two cell (cluster) types as nodes and not all cells, as we are especially interested

---

4. https://grand-challenge.org/algorithms/colon-budding-in-ihc/

Table 1: Options for the experimental setups. In total, this results in 144 setups for all combinations of graph configurations, node labels, message passing functions, and with/without JK and additional clinical information.

| Graph Options | Edge Configuration | Delaunay, Delaunay-Star, Hierarchical |
|---|---|---|
| | Node Labels | Type (tumor bud or T-cell), coordinates, ViT-16 embedding |
| GNN | MP Functions | GraphSAGE, GIN, GATv2 |
| | Jumping Knowledge (JK) | With and without |
| Additional Data | Clinical Information (SGG Criteria) | With and without |

in the interaction of the T-cells within close proximity of the tumor buds. Thus, *Delaunay* triangulation might not be the best way to capture this spatial relationship, we define two more edge insertion functions. Firstly, we propose a *Hierarchical* representation, where all T-cells nodes are connected to their closest tumor bud node, with a cut-off of $100\mu m$. Then, all T-cell nodes connected to the same tumor bud are fully-connected. Lastly, the tumor bud nodes are fully-connected as well. Secondly, we define the *Delaunay-Star* representation, the tumor bud nodes are connected using Delaunay triangulation, and the T-cell nodes are connected to all tumor bud nodes within a radius of $100\mu m$.

## 4. Experimental Evaluation

In this section, we provide the details of the experimental setups and present the results of the best-performing ones. A schematic overview of the whole experimental framework can be found in Figure 1. As a baseline, we use the risk assessment guidelines defined by the Swiss Society of Gastroentrologists (SGG) (Dieter, 2022), which define the current treatment recommendations for pT1 polyps for Swiss pathologists. For more details on the SGG criteria see Appendix A. Because the dataset is unbalanced, we use the following metrics to assess the performance: Average F1Score (average of per-class score), the True Negative Rate (TNR), and the True Positive Rate (TPR). The TNR is the recall for the low-risk class and is also called specificity or selectivity. The TPR is the recall for the high-risk group, also known as sensitivity.

### 4.1. Experimental Setup

We use a 5-fold cross-validation (CV) setup. The data is split so that patients do not overlap, and the distribution between the two classes is roughly the same in each fold (see Table D5). The evaluation is done per patient. If there are multiple graphs per patient, the classification is based on $max(softmax(\mathbf{z}_i), ..., softmax(\mathbf{z}_n))$ with $z$ being the output of the classification MLP, and $n$ the total number of graphs. Each experimental setup is run 5 times with a different seed. An ensemble of these 5 models is used for the risk classification,

Table 2: Best results for all node label and graph configuration combinations are presented here (5-model ensemble, average and standard deviation over 5-fold CV). For the complete result tables see Appendix E. As a baseline, we report the classification based on the guidelines by the SGG (see Figure 3). True Negative Rate (TNR): recall for the low-risk group. True Positive Rate (TPR): recall for the high-risk group). The top three are indicated in bold.

| Node Label | Edge Config | GNN | Clin. Info | Average F1Score (%) | TNR (%) | TPR (%) |
|---|---|---|---|---|---|---|
| Type | Delaunay | GraphSAGE-JK | Yes | $35.2_{\pm 3.3}$ | $31.4_{\pm 4.7}$ | $\mathbf{88.3_{\pm 5.1}}$ |
| | Delaunay-Star | GraphSAGE-JK | Yes | $39.5_{\pm 8.1}$ | $40.2_{\pm 13.1}$ | $79.9_{\pm 13.5}$ |
| | Hierarchical | GIN-JK | No | $30.4_{\pm 14.4}$ | $27.9_{\pm 20.7}$ | $83.7_{\pm 13.5}$ |
| Type Coord. | Delaunay | GIN-JK | No | $27.1_{\pm 11.2}$ | $24.1_{\pm 22.1}$ | $83.0_{\pm 21.4}$ |
| | Delaunay-Star | GraphSAGE-JK | Yes | $37.2_{\pm 6.6}$ | $35.9_{\pm 8.2}$ | $80.2_{\pm 5.6}$ |
| | Hierarchical | GraphSAGE | Yes | $33.5_{\pm 4.9}$ | $30.0_{\pm 8.4}$ | $84.4_{\pm 14.3}$ |
| Type ViT-16. | Delaunay | GATv2 | Yes | $36.3_{\pm 8.7}$ | $32.6_{\pm 10.8}$ | $\mathbf{90.5_{\pm 9.6}}$ |
| | Delaunay-Stars | GIN-JK | Yes | $\mathbf{40.9_{\pm 13.2}}$ | $\mathbf{41.5_{\pm 18.6}}$ | $84.2_{\pm 6.1}$ |
| | Hierarchical | GATv2 | Yes | $\mathbf{41.1_{\pm 12.7}}$ | $\mathbf{40.7_{\pm 15.6}}$ | $84.8_{\pm 8.4}$ |
| Type Coord. ViT-16 | Delaunay | GIN-JK | No | $35.2_{\pm 10.7}$ | $32.9_{\pm 14.9}$ | $83.3_{\pm 9.8}$ |
| | Delaunay-Star | GraphSAGE-JK | Yes | $39.7_{\pm 4.2}$ | $38.4_{\pm 4.8}$ | $\mathbf{85.7_{\pm 9.9}}$ |
| | Hierarchical | GraphSAGE-JK | Yes | $\mathbf{41.9_{\pm 5.4}}$ | $\mathbf{42.5_{\pm 8.9}}$ | $84.0_{\pm 8.2}$ |
| **SGG Criteria Classification** | | | | $28.2_{\pm 4.4}$ | $22.2_{\pm 4.3}$ | $85.0_{\pm 7.6}$ |

by taking the average of the softmax output. This lets us take the certainty of each model into account.

Table 1 shows the different options for the experimental setups (total of 144 experiments). The additional clinical information used as input for the classification header is based on the SGG criteria: lymphovascular invasion yes/no (one hot encoded), tumor grade (1-3), $\log_{10}(number\ of\ tumor\ buds)$ and $\log_{10}(invasion\ depth)$ in $mm$. The numerical variables are $log10$ transformed to be near normal distributed. The implementation details can be found in Appendix D and the code is available on GitHub[5].

## 4.2. Results and Discussion

Table 2 shows the best results for each graph structure and node label combination, we consider the top-3 performing models for each graph structure and node label combination in terms of average F1Score and average Recall ($\frac{TNR+TPR}{2}$) and select the one with the highest TPR among them (disregarding models with a TNR below the baseline). The complete result (with additional metrics) can be found in Appendix E. If the main goal is to prevent unnecessary surgeries, a high TNR is important. If we choose the model with the

---
5. https://github.com/digitalpathologybern/pT1-HBTG-MIDL2023

highest average F1Score, we select a model that shows an increased performance for both the TNR and TPR compared to the baseline. For both of these metrics, the GraphSAGE-JK architecture trained on the *Hierarchical-Type-Coordinate-ViT16* graphs, and using the additional clinical information, shows the best performance. If not missing any high-risk patient is of high importance, and a moderate improvement in the TNR is acceptable, the GATv2 trained on the *Delaunay-Type-ViT16* graphs, and using the additional clinical information, should be considered. It is notable that the GraphSAGE-JK model trained on the *Delaunay-Type* graphs (using the clinical data), the most simple representation, also performs well, and additionally has the lowest standard deviation. Ockham's razor would thus suggest, that this model is preferred due to its simplicity. Since this representation does not use any image-based features, it also bypasses any bias that could be caused by the image data, e.g. staining variation.

As observed in (You et al., 2020), the architecture of the GNN depends heavily on the graph. We can see this here, in that different architectures perform best for different graph structures and node label combinations. However, using a variation of *Delaunay* triangulation to create graphs seems to be a generally good choice to perform spatial analysis in digital pathology, not only when using cell graphs. Interestingly, GraphSAGE shows good performance, even though it is the simplest MP function. This is likely because larger, and more complex models are more prone to overfitting, especially on small datasets. We also observe that some combinations seem more prone to overfitting, showing a high variance between the CV-folds, especially the experiments using the GIN MP function. Using JK seems to be preferred for both GraphSAGE and GIN, but not for GATv2.

## 5. Conclusion

In this study, we explore the potential of analyzing the interaction between the tumor and the immune system at the invasive front using graph-based deep learning for risk assessment of pT1 CRC patients. We introduce our publicly available pT1 Hotspot Tumor Budding T-cell Graph (pT1-HBTG) dataset and investigate the predictive power of different graph representations (edge insertion functions and node labels). Using GNN models, especially when combined with known histopathological risk factors, we are able to outperform the risk stratification according to current treatment recommendations and decrease the number of falsely identified high-risk patients by 20%, without a decrease in sensitivity. Creating the graphs is agnostic to the way lymphocytes and tumor buds are detected and thus can be adapted to the available image processing methods. For future work, we hope to validate our proposed methods on an external cohort and extend our findings to other cancer patient groups of interest, such as stage II CRC.

## Acknowledgments

The work presented here is supported by the Rising Tide Foundation with grant number CCR-18-130. Many thanks also go to my collaborators who helped collect the patient cohort, namely Prof. Dr. med. Gerhard Seitz (Klinikum Bamberg, Germany), Prof. Dr. med. M. Vieth (Klinikum Bayreuth, Germany), Prof. Dr. med. Gieri Cathomas (formerly Kantonsspital Baselland, Switzerland) and Scott&White Healthcare (Temple, USA).

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

## Appendix A. Guidelines by the Swiss Society of Gastroentrologists

Figure 3 shows the risk assessment workflow recommended by the SGG for pT1 polyps. The criteria vary slightly between sessile and pedunculated polyps, as some of the scoring methods are different. In essence, however, they both cover tumor grade, infiltration depth, lymphovascular invasion, and tumor budding.

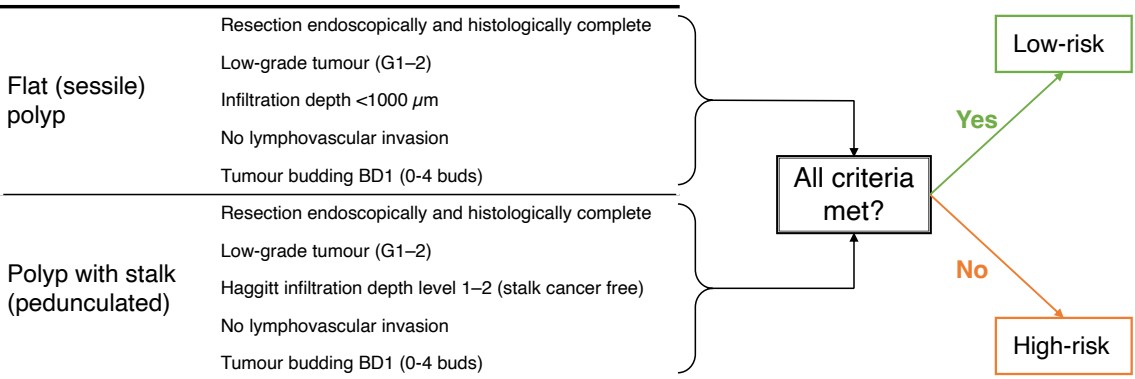

Figure 3: Prognostic classification criteria set forth by the SGG for risk assessment of pT1 polyp biopsies (Dieter, 2022). High-risk patients usually undergo colon resection, and low-risk patients follow a watch-and-wait strategy.

## Appendix B. Message-passing Functions

This section provides the mathematical details of the used MP functions.

GraphSAGE (Hamilton et al., 2017) is one of the early proposed MP functions. The hidden node state is updated as follows, using the mean aggregator scheme:

$$\mathbf{x}_i^{k+1} = \mathbf{W}_1^k \mathbf{x}_i^k + \mathbf{W}_2^k \cdot \text{mean}_{j \in \mathcal{N}(i)} \mathbf{x}_j^k, \tag{4}$$

with $\mathcal{N}(i)$ here being a fixed number of uniformly drawn neighboring nodes.

GIN (graph isomorphism network) (Xu et al., 2018a) is designed to achieve maximum discriminative power. The hidden state is updated using an MLP, $\epsilon$ is either a learnable parameter or fixed between $[0, 1]$:

$$\mathbf{x}_i^{k+1} = \text{MLP}\left((1 + \epsilon) \cdot \mathbf{x}_i^k + \sum_{j \in \mathcal{N}(i)} \mathbf{x}_j^k\right) \tag{5}$$

vh GATv2 (Brody et al., 2021) incorporates an attention mechanism into the convolution, and is able to learn neighbor-specific weights, which can not only consider node but also edge features. The attention coefficients $\alpha_{i,j}$ is defined as

$$\alpha_{ij} = softmax_j(\mathbf{a}^\top \text{LeakyReLU}(\mathbf{W} \cdot [\mathbf{x}_i \,\|\, \mathbf{x}_j \,\|\, \mathbf{e}_{i,j}]), \tag{6}$$

where $\|$ denotes vector concatenation, and $\mathbf{a} \in \mathbb{R}^{2d'}$ and $\mathbf{W} \in \mathbb{R}^{d' \times d}$ are learned. The hidden representation of node $i$ is thus updated as follows ($\sigma$ is a non-linearity):

$$\mathbf{x}_i^{k+1} = \sigma\left(\sum_{j \in \mathcal{N}(i)} \alpha_{ij}^k \cdot \mathbf{W}^k \mathbf{x}_j^k\right). \tag{7}$$

## Appendix C. Additional Details on the HBTG Dataset

Figure 4 provides more example images of the different graph types. See Table 4 for key statistics on the graphs in the different graph configuration datasets, and Table 3 for the clinical information within the high- and low-risk groups. A previous version of a subset of this dataset is described in (Eloy and Campelos, 2020). The code used to create the graph representations is available on GitHub[6]. The graph visualisation are created using a python tool that is also available on GitHub[7]

## Appendix D. Additional Details on the Experimental Setup

The implementations are all done in PyTorch (Paszke et al., 2019), specifically, PyTorch Lightning, using the PyTorch Geometric (PyG) (Fey and Lenssen, 2019) library. Table 5 shows data distribution between the CV-folds, and Table 6 shows the number of parameters for each model.

We use a setup with four MP layers with 196 neurons each, and a three-layer MLP classification header, and between each layer, we add a dropout layer ($p = 0.3$). We also

---

6. https://github.com/digitalpathologybern/BT-graph-creation

7. https://github.com/DIVA-DIA/graph_visualisation

Table 3: Statistics on the pT1 patient cohort per group of interest in regards to the clinical information.

|  | Average Invasion Depth (mm) | Lymphovascular Invasion | | Tumor Grade | | |
|  |  | Yes | No | 1 | 2 | 3 |
|---|---|---|---|---|---|---|
| Low-risk | 5.00 | 79 | 434 | 140 | 333 | 40 |
| High-risk | 6.02 | 18 | 44 | 10 | 44 | 8 |
| Total | 5.11 | 97 | 478 | 150 | 377 | 48 |

Table 4: Graph configuration overview: median of graph feature in the pT1-HBTG. The low-risk group has a median number of 171 nodes (7 for tumor buds, 157 for T-cells), and the high-risk group 193 (16 tumor buds, 180 T-cells).

|  | Delaunay | | Delaunay-Star | | Hierarchical | |
|  | Low-risk | High-risk | Low-risk | High-risk | Low-risk | High-risk |
|---|---|---|---|---|---|---|
| Edges | 496 | 561 | 63 | 192 | 224 | 622 |
| Connected Nodes | 171 | 193 | 65 | 100 | 65 | 100 |
| Isolated Nodes | 0 | 0 | 106 | 93 | 106 | 93 |

employ the GraphNorm (Cai et al., 2021) normalization implemented in PyG between each MP layer to stabilize training. GraphNorm normalizes the node representations across all nodes per graph using a learnable shift $\alpha$, which determines how much information to keep in the mean. We use the default $\alpha = 0.00001$

After the read-out phase (global add-pooling), the graph embedding vector has a size of 196 or 784 neurons, as the output of all 4 MP layers are concatenated when using JK. The second MLP layer has double as many neurons, the last one again either 196 or 784. For the configuration, where additional patient data is used as input (concatenated with the read-out vector), an additional 5 neurons are added to the first layer. LeakyReLU (Maas et al., 2013) is used as an activation function (with default $\alpha = 0.01$).

For the MP functions, we use the implementations provided by PyG, with the following configurations:

- GraphSAGE: *SAGEConv*, no additional parameters need to be specified.

- GIN: *GINConv*, with a two-layer MLP (169 neurons, $\epsilon = 0$) using the LeakyReLU activation function and dropout ($p = 0.3$). Using an MLP and an $\epsilon = 0$ is suggested by (Xu et al., 2018a).

- GATv2: *GATv2Conv*, with a single attention header (default configuration).

The models are trained for 100 epochs (until convergence), with a batch size of 256, a learning rate and weight decay of 0.0001, and the *StepLR* learning rate scheduler (step-

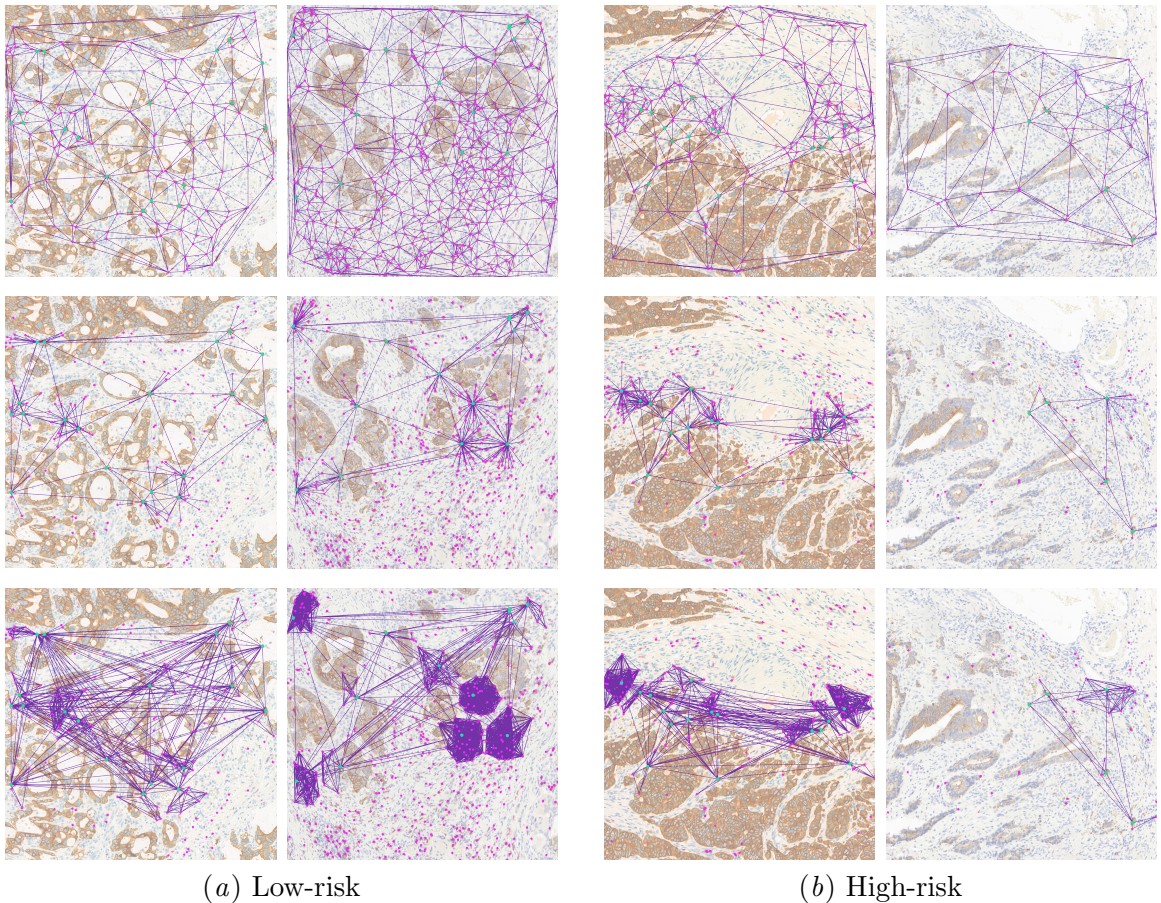

(*a*) Low-risk                    (*b*) High-risk

Figure 4: More examples from the pT1-HBTG datasets. The nodes are color-coded based on the *type* node label, tumor buds in green and T-cells in pink. The edges are in purple according to the edge insertion function described in Section 3.2.2: *Delaunay* (top), *Delaunay-Star* (middle), *Hierarchical* (bottom).

size of 10). We use the Adam optimizer and the cross-entropy loss function with label smoothing (Szegedy et al., 2016) of 0.2. As data augmentation, we employ node dropping ($p = 0.1$), where nodes are dropped randomly from the adjacency matrix with probability p using samples from a Bernoulli distribution. For the experimental setups, where the coordinates are used as node labels, we additionally add random shearing (shearing factor range $[-1.05, 1.05]$), scaling ($\pm 5\%$), and random rotation ($0-360°$) of the coordinates, before zero centering. For the coordinate augmentation, we use the transform implementations provided by the PyG library.

The design choices are made based on previous empirical experience (Studer et al., 2021b), the original publications of the MP functions, and suggestions provided by(You et al., 2020), and validated on a preliminary dataset. GraphNorm and a low learning rate increase the training stability and show a more stable loss convergence on the validation sets. Data augmentation, dropout, loss smoothing, and weight decay help with overfit-

ting, as the dataset is small and unbalanced. The number of neurons is optimized between $[128, 196, 256]$. (You et al., 2020) suggest using the Adam optimizer, 4 MP layers, concatenation for JK, and sum aggregation for read-out (global-add pooling) work best.

Table 5: Overview of the 5-fold cross-validation split. Overall, there are 626 graphs from 575 patients, as multiple slides are available for some patients. For each slide, one hotspot is annotated from which a graph is extracted.

| CV fold | # patients | | # graphs | |
|---|---|---|---|---|
| | low-risk | high-risk | low-risk | high-risk |
| 0 | 104 | 10 | 109 | 12 |
| 1 | 103 | 12 | 107 | 14 |
| 2 | 102 | 13 | 109 | 18 |
| 3 | 103 | 13 | 109 | 19 |
| 4 | 101 | 14 | 107 | 22 |
| total | 513 | 62 | 541 | 85 |

Table 6: Overview of the number of parameters in the different GNN configurations. The number depends on the message-passing (MP) function, the number of node/edge features, and whether or not Jumping Knowledge (JK) is used. The number of neurons and number of layers is kept fixed otherwise. Using the additional clinical data adds another 1,920 parameters to the respective model.

| | | Node Features | | | |
|---|---|---|---|---|---|
| | | Type | Type Coord. | Type ViT-16 | Type Coord. ViT-16 |
| # Node Features | | 2 | 2+2 | 2+384 | 2+2+384 |
| GATv2 | - | 375,938 | 376,706 | 523,394 | 524,162 |
| | JK | 2,590,466 | 2,591,234 | 2,737,922 | 2,738,690 |
| GIN | - | 411,650 | 412,034 | 485,378 | 485,762 |
| | JK | 2,626,946 | 2,627,330 | 2,700,674 | 2,701,058 |
| GraphSAGE | - | 372,866 | 373,634 | 520,322 | 521,090 |
| | JK | 2,588,162 | 2,588,930 | 2,735,618 | 2,736,386 |

## Appendix E. Full Experimental Results

In this section, we present the full results for all experimental set-ups, with the per-class F1Score and precision in addition to the per-class recall. The results considered best and presented in Table 2 are highlighted in bold.

Table 7, 10, 13, and 16 show the results for the *Delaunay* graph representation with type, type + coordinates, type + ViT-16, and type + coordinates + ViT-16, respectively.

Table 8, 11, 14, and 17 show the results for the *Delaunay-Star* graph representation with type, type + coordinates, type + ViT-16, and type + coordinates + ViT-16, respectively.

Table 9, 12, 15, and 18 show the results for the *Hierarchical* graph representation with type, type + coordinates, type + ViT-16, and type + coordinates + ViT-16, respectively.

Table 7: Full results and additional metrics of experiments on the *Delaunay* graph representations using just the node type as a node label.

| GNN | Clin. Info | F1Score Low-risk | F1Score High-risk | Precision Low-risk | Precision High-risk | Recall Low-risk (TNR) | Recall High-risk (TPR) |
|---|---|---|---|---|---|---|---|
| GATv2 | No | $23.8_{\pm 27.7}$ | $22.0_{\pm 4.4}$ | $58.3_{\pm 47.6}$ | $12.6_{\pm 3.0}$ | $16.9_{\pm 20.1}$ | $94.2_{\pm 8.4}$ |
| | Yes | $14.2_{\pm 17.5}$ | $19.9_{\pm 2.8}$ | $56.1_{\pm 46.0}$ | $11.2_{\pm 1.9}$ | $8.9_{\pm 11.4}$ | $94.1_{\pm 7.3}$ |
| GATv2-JK | No | $55.1_{\pm 10.7}$ | $23.5_{\pm 2.8}$ | $94.3_{\pm 2.5}$ | $13.9_{\pm 2.0}$ | $39.8_{\pm 10.9}$ | $79.6_{\pm 11.8}$ |
| | Yes | $54.9_{\pm 10.1}$ | $23.1_{\pm 1.7}$ | $94.9_{\pm 4.0}$ | $13.6_{\pm 0.9}$ | $39.6_{\pm 9.6}$ | $79.6_{\pm 15.3}$ |
| GIN | No | $48.6_{\pm 10.3}$ | $23.0_{\pm 3.0}$ | $94.9_{\pm 1.9}$ | $13.4_{\pm 2.1}$ | $33.4_{\pm 9.4}$ | $84.4_{\pm 9.1}$ |
| | Yes | $39.3_{\pm 16.8}$ | $21.9_{\pm 3.7}$ | $94.2_{\pm 3.6}$ | $12.6_{\pm 2.5}$ | $26.4_{\pm 13.8}$ | $86.0_{\pm 10.8}$ |
| GIN-JK | No | $57.4_{\pm 5.9}$ | $24.6_{\pm 3.2}$ | $95.1_{\pm 1.3}$ | $14.5_{\pm 2.2}$ | $41.3_{\pm 6.1}$ | $81.9_{\pm 7.3}$ |
| | Yes | $53.6_{\pm 3.0}$ | $22.7_{\pm 2.8}$ | $93.7_{\pm 1.1}$ | $13.3_{\pm 1.8}$ | $37.6_{\pm 2.9}$ | $78.7_{\pm 5.2}$ |
| GraphSAGE | No | $41.3_{\pm 8.1}$ | $22.7_{\pm 3.6}$ | $96.2_{\pm 3.2}$ | $13.1_{\pm 2.4}$ | $26.8_{\pm 7.1}$ | $90.1_{\pm 8.3}$ |
| | Yes | $32.9_{\pm 22.2}$ | $22.2_{\pm 2.9}$ | $77.8_{\pm 39.0}$ | $12.7_{\pm 2.1}$ | $22.2_{\pm 16.8}$ | $91.7_{\pm 10.2}$ |
| GraphSAGE-JK | No | $43.5_{\pm 4.5}$ | $22.7_{\pm 3.2}$ | $95.4_{\pm 1.4}$ | $13.0_{\pm 2.0}$ | $28.3_{\pm 3.8}$ | $88.4_{\pm 4.9}$ |
| | Yes | $\mathbf{47.1}_{\pm 5.6}$ | $\mathbf{23.4}_{\pm 3.0}$ | $\mathbf{95.7}_{\pm 1.9}$ | $\mathbf{13.5}_{\pm 1.9}$ | $\mathbf{31.4}_{\pm 4.7}$ | $\mathbf{88.3}_{\pm 5.1}$ |

Table 8: Full results and additional metrics of experiments on the *Delaunay-Star* graph representations using just the node type as a node label.

| GNN | Clin. Info | F1Score Low-risk | F1Score High-risk | Precision Low-risk | Precision High-risk | Recall Low-risk (TNR) | Recall High-risk (TPR) |
|---|---|---|---|---|---|---|---|
| GATv2 | No | $43.7_{\pm24.1}$ | $19.4_{\pm1.8}$ | $87.2_{\pm8.3}$ | $11.5_{\pm1.6}$ | $32.6_{\pm21.8}$ | $71.3_{\pm20.6}$ |
| | Yes | $23.8_{\pm5.4}$ | $18.7_{\pm2.1}$ | $86.6_{\pm8.6}$ | $10.5_{\pm1.2}$ | $13.8_{\pm3.5}$ | $84.3_{\pm7.3}$ |
| GATv2-JK | No | $60.4_{\pm10.8}$ | $21.5_{\pm5.5}$ | $91.9_{\pm4.1}$ | $12.9_{\pm3.4}$ | $46.0_{\pm11.5}$ | $65.8_{\pm17.9}$ |
| | Yes | $43.6_{\pm25.8}$ | $20.4_{\pm3.3}$ | $91.1_{\pm5.3}$ | $12.3_{\pm2.9}$ | $33.2_{\pm24.4}$ | $72.2_{\pm18.4}$ |
| GIN | No | $19.0_{\pm22.5}$ | $19.4_{\pm1.7}$ | $86.5_{\pm18.7}$ | $11.0_{\pm1.3}$ | $13.0_{\pm17.6}$ | $87.9_{\pm14.2}$ |
| | Yes | $28.6_{\pm20.7}$ | $21.8_{\pm3.6}$ | $95.6_{\pm6.5}$ | $12.5_{\pm2.5}$ | $19.0_{\pm16.8}$ | $92.5_{\pm9.6}$ |
| GIN-JK | No | $68.5_{\pm9.1}$ | $24.2_{\pm6.0}$ | $92.3_{\pm2.7}$ | $15.2_{\pm4.2}$ | $55.2_{\pm11.2}$ | $63.0_{\pm11.5}$ |
| | Yes | $59.2_{\pm15.9}$ | $22.5_{\pm7.2}$ | $91.6_{\pm4.2}$ | $13.6_{\pm4.7}$ | $45.4_{\pm15.0}$ | $68.1_{\pm17.7}$ |
| GraphSAGE | No | $32.6_{\pm18.7}$ | $19.9_{\pm3.4}$ | $95.4_{\pm5.8}$ | $11.4_{\pm1.9}$ | $21.7_{\pm14.5}$ | $84.0_{\pm20.6}$ |
| | Yes | $50.7_{\pm12.3}$ | $21.3_{\pm2.0}$ | $93.9_{\pm4.0}$ | $12.5_{\pm1.1}$ | $36.1_{\pm11.5}$ | $76.1_{\pm16.9}$ |
| GraphSAGE-JK | No | $56.3_{\pm11.9}$ | $22.9_{\pm3.4}$ | $92.5_{\pm1.9}$ | $13.6_{\pm2.4}$ | $41.4_{\pm12.1}$ | $73.9_{\pm3.7}$ |
| | Yes | $\mathbf{55.0_{\pm14.3}}$ | $\mathbf{24.0_{\pm4.8}}$ | $\mathbf{95.1_{\pm3.2}}$ | $\mathbf{14.2_{\pm3.2}}$ | $\mathbf{40.2_{\pm13.1}}$ | $\mathbf{79.9_{\pm13.5}}$ |

Table 9: Full results and additional metrics of experiments on the *Hierarchical* graph representations using just the node type as a node label.

| GNN | Clin. Info | F1 Score (%) | | Precision (%) | | Recall (%) | |
|---|---|---|---|---|---|---|---|
| | | Low-risk | High-risk | Low-risk | High-risk | Low-risk (TNR) | High-risk (TPR) |
| GATv2 | No | $54.2_{\pm22.3}$ | $20.9_{\pm3.2}$ | $92.6_{\pm4.0}$ | $12.9_{\pm2.4}$ | $42.7_{\pm23.3}$ | $68.5_{\pm23.9}$ |
| | Yes | $37.0_{\pm26.9}$ | $18.8_{\pm6.9}$ | $70.6_{\pm35.6}$ | $11.0_{\pm4.4}$ | $27.5_{\pm22.4}$ | $72.4_{\pm26.8}$ |
| GATv2-JK | No | $34.4_{\pm18.3}$ | $18.5_{\pm3.6}$ | $89.0_{\pm10.0}$ | $10.6_{\pm2.2}$ | $23.0_{\pm13.8}$ | $75.7_{\pm18.4}$ |
| | Yes | $37.7_{\pm6.9}$ | $17.1_{\pm3.1}$ | $86.7_{\pm6.2}$ | $9.8_{\pm1.8}$ | $24.4_{\pm5.6}$ | $68.9_{\pm15.0}$ |
| GIN | No | $2.3_{\pm2.2}$ | $19.0_{\pm1.4}$ | $50.0_{\pm44.7}$ | $10.6_{\pm0.8}$ | $1.2_{\pm1.1}$ | $97.1_{\pm5.7}$ |
| | Yes | $1.8_{\pm3.7}$ | $19.6_{\pm1.6}$ | $20.0_{\pm40.0}$ | $10.9_{\pm1.0}$ | $1.0_{\pm1.9}$ | $100.0_{\pm0.0}$ |
| GIN-JK | No | $\mathbf{38.8_{\pm25.6}}$ | $\mathbf{22.1_{\pm4.7}}$ | $\mathbf{91.4_{\pm11.1}}$ | $12.9_{\pm3.4}$ | $\mathbf{27.9_{\pm20.7}}$ | $\mathbf{83.7_{\pm13.5}}$ |
| | Yes | $48.4_{\pm20.5}$ | $21.2_{\pm4.7}$ | $88.9_{\pm8.3}$ | $12.6_{\pm3.1}$ | $35.4_{\pm17.4}$ | $73.6_{\pm15.3}$ |
| GraphSAGE | No | $36.0_{\pm20.7}$ | $21.1_{\pm3.3}$ | $94.2_{\pm3.3}$ | $12.2_{\pm2.2}$ | $24.8_{\pm17.7}$ | $84.7_{\pm16.1}$ |
| | Yes | $46.0_{\pm22.3}$ | $22.5_{\pm5.2}$ | $91.9_{\pm3.3}$ | $13.3_{\pm3.8}$ | $33.6_{\pm19.6}$ | $78.9_{\pm10.7}$ |
| GraphSAGE-JK | No | $51.2_{\pm16.1}$ | $21.6_{\pm4.0}$ | $92.9_{\pm4.4}$ | $12.8_{\pm2.6}$ | $37.2_{\pm15.3}$ | $74.3_{\pm16.0}$ |
| | Yes | $63.7_{\pm4.7}$ | $23.7_{\pm4.0}$ | $93.6_{\pm3.2}$ | $14.3_{\pm2.6}$ | $48.6_{\pm6.1}$ | $71.5_{\pm14.8}$ |

Table 10: Full results and additional metrics of experiments on the *Delaunay* graph representations using the node type and the coordinates as a node label.

| GNN | Clin. Info | F1 Score (%) | | Precision (%) | | Recall (%) | |
| --- | --- | --- | --- | --- | --- | --- | --- |
| | | Low-risk | High-risk | Low-risk | High-risk | Low-risk (TNR) | High-risk (TPR) |
| GATv2 | No | $13.7_{\pm 7.3}$ | $19.5_{\pm 2.4}$ | $91.0_{\pm 7.6}$ | $10.9_{\pm 1.4}$ | $7.6_{\pm 4.4}$ | $93.5_{\pm 5.9}$ |
| | Yes | $22.7_{\pm 13.7}$ | $20.1_{\pm 2.8}$ | $73.0_{\pm 36.7}$ | $11.3_{\pm 1.8}$ | $13.7_{\pm 8.7}$ | $90.4_{\pm 5.7}$ |
| GATv2-JK | No | $29.8_{\pm 21.8}$ | $17.4_{\pm 3.7}$ | $86.6_{\pm 11.7}$ | $9.9_{\pm 2.0}$ | $20.5_{\pm 16.7}$ | $75.9_{\pm 25.7}$ |
| | Yes | $26.0_{\pm 16.4}$ | $19.9_{\pm 1.4}$ | $93.9_{\pm 5.3}$ | $11.3_{\pm 1.0}$ | $16.4_{\pm 11.9}$ | $88.1_{\pm 13.2}$ |
| GIN | No | $18.2_{\pm 12.0}$ | $19.8_{\pm 1.5}$ | $95.0_{\pm 6.1}$ | $11.1_{\pm 1.0}$ | $10.8_{\pm 8.5}$ | $92.7_{\pm 11.1}$ |
| | Yes | $16.2_{\pm 6.2}$ | $20.4_{\pm 2.7}$ | $97.1_{\pm 5.7}$ | $11.4_{\pm 1.7}$ | $9.0_{\pm 3.7}$ | $96.7_{\pm 6.7}$ |
| GIN-JK | No | $\mathbf{34.0}_{\pm 22.4}$ | $\mathbf{20.3}_{\pm 1.9}$ | $\mathbf{95.2}_{\pm 4.3}$ | $\mathbf{11.8}_{\pm 1.6}$ | $\mathbf{24.1}_{\pm 21.1}$ | $\mathbf{83.0}_{\pm 21.4}$ |
| | Yes | $27.9_{\pm 18.5}$ | $19.2_{\pm 3.1}$ | $89.3_{\pm 6.9}$ | $11.0_{\pm 2.0}$ | $18.2_{\pm 13.8}$ | $82.2_{\pm 13.4}$ |
| GraphSAGE | No | $19.1_{\pm 12.8}$ | $19.3_{\pm 2.2}$ | $89.4_{\pm 6.7}$ | $10.9_{\pm 1.4}$ | $11.3_{\pm 8.4}$ | $89.1_{\pm 7.2}$ |
| | Yes | $16.5_{\pm 7.9}$ | $18.9_{\pm 2.5}$ | $90.7_{\pm 7.8}$ | $10.6_{\pm 1.5}$ | $9.4_{\pm 4.9}$ | $88.6_{\pm 9.5}$ |
| GraphSAGE-JK | No | $21.2_{\pm 13.8}$ | $20.5_{\pm 2.8}$ | $95.8_{\pm 3.6}$ | $11.6_{\pm 1.8}$ | $12.7_{\pm 9.3}$ | $93.6_{\pm 5.6}$ |
| | Yes | $25.1_{\pm 12.8}$ | $17.3_{\pm 3.4}$ | $87.9_{\pm 8.5}$ | $9.8_{\pm 1.8}$ | $15.6_{\pm 9.7}$ | $78.5_{\pm 22.5}$ |

Table 11: Full results and additional metrics of experiments on the *Delaunay-Star* graph representations using the node type and the coordinates as a node label.

| GNN | Clin. Info | F1 Score (%) | | Precision (%) | | Recall (%) | |
| --- | --- | --- | --- | --- | --- | --- | --- |
| | | Low-risk | High-risk | Low-risk | High-risk | Low-risk (TNR) | High-risk (TPR) |
| GATv2 | No | $19.5_{\pm 9.2}$ | $18.9_{\pm 2.2}$ | $88.3_{\pm 6.0}$ | $10.6_{\pm 1.3}$ | $11.3_{\pm 6.2}$ | $87.2_{\pm 10.2}$ |
| | Yes | $17.7_{\pm 6.6}$ | $19.1_{\pm 3.2}$ | $86.4_{\pm 16.8}$ | $10.7_{\pm 1.9}$ | $9.9_{\pm 4.0}$ | $89.3_{\pm 13.1}$ |
| GATv2-JK | No | $49.4_{\pm 7.8}$ | $21.6_{\pm 2.9}$ | $94.1_{\pm 4.2}$ | $12.6_{\pm 1.7}$ | $34.1_{\pm 7.5}$ | $79.2_{\pm 15.8}$ |
| | Yes | $38.0_{\pm 5.6}$ | $21.4_{\pm 2.0}$ | $93.8_{\pm 1.9}$ | $12.2_{\pm 1.3}$ | $24.0_{\pm 4.6}$ | $87.3_{\pm 2.9}$ |
| GIN | No | $21.7_{\pm 26.6}$ | $20.5_{\pm 3.3}$ | $56.6_{\pm 46.3}$ | $11.7_{\pm 2.4}$ | $15.5_{\pm 19.7}$ | $89.7_{\pm 12.8}$ |
| | Yes | $14.8_{\pm 23.5}$ | $18.8_{\pm 1.4}$ | $52.8_{\pm 43.8}$ | $10.6_{\pm 0.9}$ | $10.7_{\pm 18.1}$ | $87.9_{\pm 17.9}$ |
| GIN-JK | No | $35.7_{\pm 25.0}$ | $20.6_{\pm 2.6}$ | $94.9_{\pm 4.5}$ | $12.0_{\pm 1.9}$ | $25.5_{\pm 19.2}$ | $82.2_{\pm 16.9}$ |
| | Yes | $45.8_{\pm 31.7}$ | $23.2_{\pm 6.2}$ | $92.4_{\pm 5.7}$ | $14.2_{\pm 4.7}$ | $36.7_{\pm 27.6}$ | $77.1_{\pm 18.4}$ |
| GraphSAGE | No | $45.1_{\pm 9.3}$ | $21.8_{\pm 3.7}$ | $93.2_{\pm 3.1}$ | $12.6_{\pm 2.4}$ | $30.2_{\pm 8.0}$ | $82.4_{\pm 8.7}$ |
| | Yes | $42.1_{\pm 8.6}$ | $19.9_{\pm 3.7}$ | $91.4_{\pm 3.8}$ | $11.5_{\pm 2.2}$ | $27.9_{\pm 7.5}$ | $76.9_{\pm 12.9}$ |
| GraphSAGE-JK | No | $54.6_{\pm 7.1}$ | $22.9_{\pm 4.9}$ | $93.1_{\pm 2.8}$ | $13.5_{\pm 3.1}$ | $38.8_{\pm 6.7}$ | $77.0_{\pm 6.7}$ |
| | Yes | $\mathbf{51.4}_{\pm 9.1}$ | $\mathbf{22.9}_{\pm 4.4}$ | $\mathbf{93.5}_{\pm 2.3}$ | $\mathbf{13.4}_{\pm 2.9}$ | $\mathbf{35.9}_{\pm 8.2}$ | $\mathbf{80.2}_{\pm 5.6}$ |

Table 12: Full results and additional metrics of experiments on the *Hierarchical* graph representations using the node type and the coordinates as a node label.

| GNN | Clin. Info | F1 Score (%) | | Precision (%) | | Recall (%) | |
| --- | --- | --- | --- | --- | --- | --- | --- |
| | | Low-risk | High-risk | Low-risk | High-risk | Low-risk (TNR) | High-risk (TPR) |
| GATv2 | No | $26.7_{\pm 12.1}$ | $20.5_{\pm 2.0}$ | $95.4_{\pm 4.4}$ | $11.6_{\pm 1.3}$ | $16.3_{\pm 8.4}$ | $91.2_{\pm 10.5}$ |
| | Yes | $22.8_{\pm 7.4}$ | $19.2_{\pm 1.7}$ | $92.2_{\pm 7.3}$ | $10.8_{\pm 1.0}$ | $13.3_{\pm 5.0}$ | $88.0_{\pm 11.8}$ |
| GATv2-JK | No | $40.4_{\pm 3.3}$ | $20.3_{\pm 3.2}$ | $91.9_{\pm 3.6}$ | $11.6_{\pm 2.0}$ | $26.0_{\pm 2.8}$ | $80.5_{\pm 9.7}$ |
| | Yes | $39.3_{\pm 17.8}$ | $21.7_{\pm 3.4}$ | $95.7_{\pm 4.7}$ | $12.5_{\pm 2.2}$ | $26.5_{\pm 13.9}$ | $86.0_{\pm 14.9}$ |
| GIN | No | $0.4_{\pm 0.8}$ | $19.2_{\pm 1.9}$ | $20.0_{\pm 40.0}$ | $10.6_{\pm 1.1}$ | $0.2_{\pm 0.4}$ | $98.5_{\pm 3.1}$ |
| | Yes | $0.8_{\pm 1.0}$ | $19.5_{\pm 1.9}$ | $40.0_{\pm 49.0}$ | $10.8_{\pm 1.2}$ | $0.4_{\pm 0.5}$ | $100.0_{\pm 0.0}$ |
| GIN-JK | No | $19.2_{\pm 23.8}$ | $20.2_{\pm 4.0}$ | $80.9_{\pm 15.5}$ | $11.5_{\pm 2.8}$ | $13.4_{\pm 19.1}$ | $88.9_{\pm 7.8}$ |
| | Yes | $23.0_{\pm 17.1}$ | $19.7_{\pm 2.5}$ | $71.6_{\pm 36.0}$ | $11.1_{\pm 1.5}$ | $14.4_{\pm 11.5}$ | $88.1_{\pm 10.1}$ |
| GraphSAGE | No | $48.3_{\pm 16.9}$ | $22.1_{\pm 4.1}$ | $92.8_{\pm 1.2}$ | $13.0_{\pm 2.9}$ | $34.6_{\pm 15.8}$ | $77.8_{\pm 10.5}$ |
| | Yes | $\mathbf{44.9_{\pm 9.2}}$ | $\mathbf{22.2_{\pm 4.1}}$ | $\mathbf{95.0_{\pm 3.4}}$ | $\mathbf{12.8_{\pm 2.5}}$ | $\mathbf{30.0_{\pm 8.4}}$ | $\mathbf{84.4_{\pm 14.3}}$ |
| GraphSAGE-JK | No | $53.2_{\pm 9.6}$ | $23.0_{\pm 3.5}$ | $93.9_{\pm 2.0}$ | $13.6_{\pm 2.4}$ | $37.9_{\pm 9.6}$ | $79.3_{\pm 8.4}$ |
| | Yes | $57.0_{\pm 7.4}$ | $23.5_{\pm 3.2}$ | $94.0_{\pm 2.6}$ | $13.9_{\pm 2.1}$ | $41.4_{\pm 7.8}$ | $78.1_{\pm 11.6}$ |

Table 13: Full results and additional metrics of experiments on the *Delaunay* graph representations using the node type and the ViT-16 embedding as a node label.

| GNN | Clin. Info | F1 Score (%) | | Precision (%) | | Recall (%) | |
| --- | --- | --- | --- | --- | --- | --- | --- |
| | | Low-risk | High-risk | Low-risk | High-risk | Low-risk (TNR) | High-risk (TPR) |
| GATv2 | No | $51.2_{\pm 7.6}$ | $24.7_{\pm 6.1}$ | $95.8_{\pm 4.0}$ | $14.4_{\pm 3.8}$ | $35.1_{\pm 6.5}$ | $87.6_{\pm 11.4}$ |
| | Yes | $\mathbf{47.9_{\pm 11.6}}$ | $\mathbf{24.7_{\pm 5.9}}$ | $\mathbf{96.4_{\pm 3.9}}$ | $\mathbf{14.4_{\pm 4.0}}$ | $\mathbf{32.6_{\pm 10.8}}$ | $\mathbf{90.5_{\pm 9.6}}$ |
| GATv2-JK | No | $60.1_{\pm 8.1}$ | $24.1_{\pm 5.8}$ | $93.7_{\pm 2.3}$ | $14.5_{\pm 3.9}$ | $44.7_{\pm 8.7}$ | $75.3_{\pm 10.3}$ |
| | Yes | $57.7_{\pm 8.7}$ | $24.3_{\pm 5.7}$ | $94.2_{\pm 2.4}$ | $14.5_{\pm 3.8}$ | $42.2_{\pm 8.8}$ | $78.3_{\pm 9.6}$ |
| GIN | No | $53.1_{\pm 9.1}$ | $22.8_{\pm 5.4}$ | $93.6_{\pm 2.7}$ | $13.4_{\pm 3.4}$ | $37.7_{\pm 8.8}$ | $78.1_{\pm 12.3}$ |
| | Yes | $51.7_{\pm 2.7}$ | $24.2_{\pm 2.6}$ | $95.8_{\pm 1.2}$ | $14.1_{\pm 1.7}$ | $35.5_{\pm 2.6}$ | $87.2_{\pm 3.4}$ |
| GIN-JK | No | $68.1_{\pm 4.7}$ | $25.6_{\pm 5.4}$ | $93.9_{\pm 1.5}$ | $15.7_{\pm 3.7}$ | $53.6_{\pm 5.8}$ | $70.5_{\pm 10.7}$ |
| | Yes | $67.1_{\pm 6.1}$ | $25.0_{\pm 3.6}$ | $93.6_{\pm 1.4}$ | $15.4_{\pm 2.7}$ | $52.7_{\pm 7.4}$ | $69.9_{\pm 7.3}$ |
| GraphSAGE | No | $63.5_{\pm 7.8}$ | $25.1_{\pm 4.7}$ | $93.9_{\pm 0.6}$ | $15.2_{\pm 3.4}$ | $48.5_{\pm 9.2}$ | $73.9_{\pm 3.7}$ |
| | Yes | $58.7_{\pm 10.1}$ | $22.4_{\pm 6.0}$ | $91.7_{\pm 2.5}$ | $13.5_{\pm 4.0}$ | $43.7_{\pm 10.1}$ | $68.7_{\pm 6.0}$ |
| GraphSAGE-JK | No | $64.0_{\pm 12.8}$ | $27.7_{\pm 8.0}$ | $94.8_{\pm 1.7}$ | $17.1_{\pm 5.8}$ | $49.6_{\pm 14.2}$ | $78.5_{\pm 5.9}$ |
| | Yes | $65.1_{\pm 5.6}$ | $25.0_{\pm 6.3}$ | $93.7_{\pm 1.8}$ | $15.3_{\pm 4.4}$ | $50.1_{\pm 6.3}$ | $72.0_{\pm 9.1}$ |

Table 14: Full results and additional metrics of experiments on the *Delaunay-Star* graph representations using the node type and the ViT-16 embedding as a node label.

| GNN | Clin. Info | F1 Score (%) | | Precision (%) | | Recall (%) | |
|-----|------|----------|-----------|----------|-----------|----------------------|----------------------|
| | | Low-risk | High-risk | Low-risk | High-risk | Low-risk (TNR) | High-risk (TPR) |
| GATv2 | No | $42.9_{\pm 8.1}$ | $20.8_{\pm 5.2}$ | $92.1_{\pm 4.2}$ | $12.0_{\pm 3.1}$ | $28.3_{\pm 6.3}$ | $79.3_{\pm 13.5}$ |
| | Yes | $51.4_{\pm 9.0}$ | $21.6_{\pm 4.6}$ | $92.2_{\pm 2.6}$ | $12.7_{\pm 3.0}$ | $36.1_{\pm 8.4}$ | $75.2_{\pm 8.2}$ |
| GATv2-JK | No | $61.6_{\pm 6.6}$ | $23.8_{\pm 7.2}$ | $93.0_{\pm 3.0}$ | $14.4_{\pm 4.8}$ | $46.2_{\pm 6.8}$ | $71.6_{\pm 11.7}$ |
| | Yes | $63.9_{\pm 12.3}$ | $27.6_{\pm 8.9}$ | $94.4_{\pm 3.2}$ | $17.0_{\pm 6.2}$ | $49.2_{\pm 13.1}$ | $78.3_{\pm 8.2}$ |
| GIN | No | $37.1_{\pm 27.0}$ | $23.4_{\pm 7.6}$ | $91.4_{\pm 3.6}$ | $14.0_{\pm 5.6}$ | $27.3_{\pm 23.9}$ | $85.9_{\pm 6.7}$ |
| | Yes | $45.8_{\pm 27.6}$ | $24.0_{\pm 5.7}$ | $95.1_{\pm 2.6}$ | $14.3_{\pm 4.1}$ | $34.7_{\pm 23.0}$ | $82.9_{\pm 10.8}$ |
| GIN-JK | No | $59.0_{\pm 21.6}$ | $26.4_{\pm 9.3}$ | $92.5_{\pm 3.9}$ | $16.4_{\pm 6.9}$ | $46.3_{\pm 20.9}$ | $75.2_{\pm 9.3}$ |
| | Yes | $\mathbf{55.3}_{\pm \mathbf{19.4}}$ | $\mathbf{26.5}_{\pm \mathbf{7.0}}$ | $\mathbf{95.0}_{\pm \mathbf{2.0}}$ | $\mathbf{16.0}_{\pm \mathbf{5.0}}$ | $\mathbf{41.5}_{\pm \mathbf{18.6}}$ | $\mathbf{84.2}_{\pm \mathbf{6.1}}$ |
| GraphSAGE | No | $58.1_{\pm 9.6}$ | $25.6_{\pm 7.8}$ | $94.6_{\pm 3.6}$ | $15.3_{\pm 5.3}$ | $42.4_{\pm 9.8}$ | $81.3_{\pm 11.0}$ |
| | Yes | $59.4_{\pm 9.6}$ | $24.8_{\pm 8.2}$ | $94.0_{\pm 4.1}$ | $14.9_{\pm 5.5}$ | $44.1_{\pm 10.6}$ | $76.6_{\pm 16.5}$ |
| GraphSAGE-JK | No | $63.4_{\pm 7.6}$ | $25.5_{\pm 6.8}$ | $94.0_{\pm 2.6}$ | $15.5_{\pm 4.6}$ | $48.2_{\pm 8.1}$ | $75.1_{\pm 9.5}$ |
| | Yes | $66.7_{\pm 6.5}$ | $27.1_{\pm 6.1}$ | $94.9_{\pm 1.6}$ | $16.6_{\pm 4.1}$ | $51.7_{\pm 7.2}$ | $76.8_{\pm 7.6}$ |

Table 15: Full results and additional metrics of experiments on the *Hierarchical* graph representations using the node type and the ViT-16 embedding as a node label.

| GNN | Clin. Info | F1 Score (%) | | Precision (%) | | Recall (%) | |
|-----|------|----------|-----------|----------|-----------|----------------------|----------------------|
| | | Low-risk | High-risk | Low-risk | High-risk | Low-risk (TNR) | High-risk (TPR) |
| GATv2 | No | $44.4_{\pm 11.2}$ | $22.7_{\pm 6.1}$ | $93.9_{\pm 4.5}$ | $13.2_{\pm 4.0}$ | $29.7_{\pm 10.2}$ | $85.0_{\pm 9.8}$ |
| | Yes | $\mathbf{55.5}_{\pm \mathbf{15.7}}$ | $\mathbf{26.8}_{\pm \mathbf{10.0}}$ | $\mathbf{94.7}_{\pm \mathbf{3.6}}$ | $\mathbf{16.2}_{\pm \mathbf{6.9}}$ | $\mathbf{40.7}_{\pm \mathbf{15.6}}$ | $\mathbf{84.8}_{\pm \mathbf{8.4}}$ |
| GATv2-JK | No | $57.1_{\pm 7.0}$ | $23.6_{\pm 5.5}$ | $93.6_{\pm 2.2}$ | $14.0_{\pm 3.6}$ | $41.4_{\pm 7.1}$ | $76.8_{\pm 7.6}$ |
| | Yes | $58.4_{\pm 8.6}$ | $23.0_{\pm 6.7}$ | $92.6_{\pm 3.8}$ | $13.8_{\pm 4.4}$ | $43.2_{\pm 8.8}$ | $72.4_{\pm 12.4}$ |
| GIN | No | $5.5_{\pm 6.0}$ | $19.3_{\pm 1.3}$ | $56.0_{\pm 46.3}$ | $10.7_{\pm 0.8}$ | $2.9_{\pm 3.3}$ | $97.1_{\pm 5.7}$ |
| | Yes | $5.4_{\pm 7.1}$ | $19.7_{\pm 2.2}$ | $80.0_{\pm 40.0}$ | $10.9_{\pm 1.3}$ | $2.9_{\pm 3.9}$ | $98.5_{\pm 3.1}$ |
| GIN-JK | No | $59.3_{\pm 3.5}$ | $22.2_{\pm 5.0}$ | $92.6_{\pm 2.1}$ | $13.2_{\pm 3.2}$ | $43.7_{\pm 3.7}$ | $70.0_{\pm 10.3}$ |
| | Yes | $57.2_{\pm 13.7}$ | $23.3_{\pm 5.4}$ | $92.8_{\pm 4.6}$ | $14.0_{\pm 3.6}$ | $42.6_{\pm 14.2}$ | $73.5_{\pm 13.3}$ |
| GraphSAGE | No | $63.6_{\pm 6.9}$ | $25.0_{\pm 7.1}$ | $93.8_{\pm 3.2}$ | $15.1_{\pm 4.6}$ | $48.4_{\pm 7.2}$ | $73.6_{\pm 13.3}$ |
| | Yes | $58.8_{\pm 9.1}$ | $25.2_{\pm 6.5}$ | $94.4_{\pm 2.6}$ | $15.0_{\pm 4.3}$ | $43.1_{\pm 8.8}$ | $79.9_{\pm 8.0}$ |
| GraphSAGE-JK | No | $61.9_{\pm 11.7}$ | $25.1_{\pm 7.1}$ | $93.3_{\pm 2.7}$ | $15.3_{\pm 4.8}$ | $47.3_{\pm 12.2}$ | $73.7_{\pm 6.9}$ |
| | Yes | $61.0_{\pm 9.8}$ | $25.8_{\pm 7.1}$ | $94.4_{\pm 2.3}$ | $15.5_{\pm 4.8}$ | $45.7_{\pm 10.3}$ | $78.3_{\pm 7.7}$ |

Table 16: Full results and additional metrics of experiments on the *Delaunay* graph representations using the node type, the coordinates and the ViT-16 embedding as a node label.

| GNN | Clin. Info | F1 Score (%) | | Precision (%) | | Recall (%) | |
|---|---|---|---|---|---|---|---|
| | | Low-risk | High-risk | Low-risk | High-risk | Low-risk (TNR) | High-risk (TPR) |
| GATv2 | No | $24.5_{\pm 11.3}$ | $19.8_{\pm 2.2}$ | $89.7_{\pm 8.2}$ | $11.2_{\pm 1.4}$ | $14.7_{\pm 7.5}$ | $88.8_{\pm 7.4}$ |
| | Yes | $33.1_{\pm 13.4}$ | $20.4_{\pm 3.4}$ | $90.6_{\pm 3.4}$ | $11.7_{\pm 2.3}$ | $21.1_{\pm 11.1}$ | $84.2_{\pm 3.6}$ |
| GATv2-JK | No | $49.2_{\pm 15.7}$ | $22.2_{\pm 6.2}$ | $91.5_{\pm 4.0}$ | $13.1_{\pm 4.2}$ | $34.9_{\pm 14.0}$ | $76.8_{\pm 7.6}$ |
| | Yes | $46.9_{\pm 16.1}$ | $22.9_{\pm 6.0}$ | $92.7_{\pm 3.0}$ | $13.4_{\pm 4.1}$ | $32.8_{\pm 14.5}$ | $81.7_{\pm 5.9}$ |
| GIN | No | $36.6_{\pm 4.5}$ | $21.5_{\pm 2.6}$ | $94.3_{\pm 1.8}$ | $12.2_{\pm 1.6}$ | $22.8_{\pm 3.4}$ | $88.7_{\pm 3.6}$ |
| | Yes | $30.6_{\pm 10.0}$ | $20.7_{\pm 3.1}$ | $92.1_{\pm 5.1}$ | $11.8_{\pm 2.0}$ | $18.8_{\pm 7.4}$ | $88.7_{\pm 3.6}$ |
| GIN-JK | No | $\mathbf{47.0}_{\pm 15.5}$ | $\mathbf{23.4}_{\pm 6.4}$ | $\mathbf{93.9}_{\pm 4.1}$ | $\mathbf{13.8}_{\pm 4.4}$ | $\mathbf{32.9}_{\pm 14.9}$ | $\mathbf{83.3}_{\pm 9.8}$ |
| | Yes | $45.0_{\pm 11.1}$ | $21.5_{\pm 4.1}$ | $92.5_{\pm 1.3}$ | $12.5_{\pm 2.7}$ | $30.5_{\pm 10.0}$ | $80.2_{\pm 5.6}$ |
| GraphSAGE | No | $38.1_{\pm 5.6}$ | $21.4_{\pm 3.6}$ | $93.6_{\pm 2.6}$ | $12.2_{\pm 2.3}$ | $24.0_{\pm 4.3}$ | $86.6_{\pm 5.1}$ |
| | Yes | $51.6_{\pm 14.4}$ | $24.3_{\pm 9.6}$ | $92.9_{\pm 4.5}$ | $14.6_{\pm 6.7}$ | $37.0_{\pm 14.8}$ | $80.2_{\pm 11.0}$ |
| GraphSAGE-JK | No | $47.1_{\pm 15.1}$ | $21.7_{\pm 6.7}$ | $90.5_{\pm 4.3}$ | $12.8_{\pm 4.6}$ | $33.1_{\pm 14.4}$ | $75.4_{\pm 6.4}$ |
| | Yes | $47.9_{\pm 14.7}$ | $22.8_{\pm 5.7}$ | $92.6_{\pm 3.3}$ | $13.5_{\pm 4.1}$ | $33.6_{\pm 14.2}$ | $80.6_{\pm 3.7}$ |

Table 17: Full results and additional metrics of experiments on the *Delaunay-Star* graph representations using the node type, the coordinates and the ViT-16 embedding as a node label.

| GNN | Clin. Info | F1 Score (%) | | Precision (%) | | Recall (%) | |
|---|---|---|---|---|---|---|---|
| | | Low-risk | High-risk | Low-risk | High-risk | Low-risk (TNR) | High-risk (TPR) |
| GATv2 | No | $33.3_{\pm 16.7}$ | $21.2_{\pm 2.6}$ | $95.2_{\pm 3.9}$ | $12.1_{\pm 1.9}$ | $21.8_{\pm 13.4}$ | $88.1_{\pm 10.8}$ |
| | Yes | $31.7_{\pm 6.5}$ | $20.7_{\pm 3.0}$ | $93.3_{\pm 1.9}$ | $11.8_{\pm 1.9}$ | $19.3_{\pm 4.6}$ | $88.3_{\pm 5.1}$ |
| GATv2-JK | No | $49.5_{\pm 12.2}$ | $23.4_{\pm 5.8}$ | $93.8_{\pm 2.5}$ | $13.8_{\pm 4.0}$ | $34.6_{\pm 12.0}$ | $82.5_{\pm 5.6}$ |
| | Yes | $50.1_{\pm 12.3}$ | $22.9_{\pm 3.6}$ | $94.3_{\pm 3.4}$ | $13.4_{\pm 2.5}$ | $35.2_{\pm 11.3}$ | $81.6_{\pm 10.3}$ |
| GIN | No | $18.7_{\pm 23.7}$ | $20.9_{\pm 3.5}$ | $57.3_{\pm 46.9}$ | $11.8_{\pm 2.4}$ | $12.7_{\pm 16.9}$ | $93.8_{\pm 7.5}$ |
| | Yes | $4.5_{\pm 2.5}$ | $19.8_{\pm 1.8}$ | $80.0_{\pm 40.0}$ | $11.0_{\pm 1.1}$ | $2.3_{\pm 1.3}$ | $100.0_{\pm 0.0}$ |
| GIN-JK | No | $51.6_{\pm 24.2}$ | $21.8_{\pm 5.0}$ | $88.1_{\pm 8.4}$ | $13.2_{\pm 3.8}$ | $39.7_{\pm 21.4}$ | $70.1_{\pm 10.7}$ |
| | Yes | $47.4_{\pm 20.6}$ | $23.9_{\pm 4.9}$ | $95.3_{\pm 3.1}$ | $14.2_{\pm 3.5}$ | $34.4_{\pm 18.8}$ | $84.5_{\pm 10.4}$ |
| GraphSAGE | No | $43.5_{\pm 9.5}$ | $22.5_{\pm 4.1}$ | $94.7_{\pm 4.1}$ | $13.0_{\pm 2.5}$ | $28.7_{\pm 7.8}$ | $86.9_{\pm 10.9}$ |
| | Yes | $44.0_{\pm 5.4}$ | $22.8_{\pm 4.0}$ | $95.2_{\pm 2.2}$ | $13.1_{\pm 2.5}$ | $28.7_{\pm 4.3}$ | $88.2_{\pm 5.4}$ |
| GraphSAGE-JK | No | $53.5_{\pm 9.1}$ | $23.6_{\pm 5.7}$ | $93.8_{\pm 2.0}$ | $13.9_{\pm 3.8}$ | $37.9_{\pm 8.9}$ | $80.1_{\pm 5.9}$ |
| | Yes | $\mathbf{54.7}_{\pm 5.0}$ | $\mathbf{24.7}_{\pm 3.7}$ | $\mathbf{95.7}_{\pm 2.9}$ | $\mathbf{14.5}_{\pm 2.3}$ | $\mathbf{38.4}_{\pm 4.8}$ | $\mathbf{85.7}_{\pm 9.9}$ |

Table 18: Full results and additional metrics of experiments on the *Hierarchical* graph representations using the node type, the coordinates and the ViT-16 embedding as a node label.

| GNN | Clin. Info | F1 Score (%) | | Precision (%) | | Recall (%) | |
|---|---|---|---|---|---|---|---|
| | | Low-risk | High-risk | Low-risk | High-risk | Low-risk (TNR) | High-risk (TPR) |
| GATv2 | No | $25.9_{\pm8.1}$ | $21.0_{\pm3.3}$ | $94.7_{\pm7.2}$ | $11.8_{\pm2.0}$ | $15.3_{\pm5.5}$ | $93.6_{\pm7.9}$ |
| | Yes | $27.2_{\pm12.9}$ | $21.3_{\pm2.7}$ | $96.2_{\pm5.2}$ | $12.0_{\pm1.7}$ | $16.6_{\pm9.3}$ | $93.8_{\pm7.5}$ |
| GATv2-JK | No | $45.5_{\pm14.3}$ | $22.0_{\pm4.4}$ | $91.8_{\pm3.6}$ | $12.8_{\pm2.8}$ | $31.2_{\pm11.7}$ | $80.7_{\pm2.9}$ |
| | Yes | $48.6_{\pm16.2}$ | $22.5_{\pm4.4}$ | $92.3_{\pm3.6}$ | $13.2_{\pm3.0}$ | $34.4_{\pm14.0}$ | $79.4_{\pm6.7}$ |
| GIN | No | $1.2_{\pm0.9}$ | $19.5_{\pm1.9}$ | $60.0_{\pm49.0}$ | $10.8_{\pm1.1}$ | $0.6_{\pm0.5}$ | $100.0_{\pm0.0}$ |
| | Yes | $0.4_{\pm0.8}$ | $19.5_{\pm1.8}$ | $20.0_{\pm40.0}$ | $10.8_{\pm1.1}$ | $0.2_{\pm0.4}$ | $100.0_{\pm0.0}$ |
| GIN-JK | No | $34.1_{\pm22.2}$ | $21.4_{\pm4.1}$ | $93.8_{\pm4.0}$ | $12.4_{\pm2.8}$ | $23.4_{\pm17.9}$ | $85.5_{\pm10.5}$ |
| | Yes | $43.2_{\pm21.1}$ | $22.3_{\pm4.8}$ | $92.2_{\pm3.4}$ | $13.2_{\pm3.5}$ | $30.9_{\pm19.2}$ | $81.3_{\pm9.4}$ |
| GraphSAGE | No | $38.0_{\pm15.2}$ | $21.9_{\pm5.0}$ | $91.3_{\pm7.2}$ | $12.6_{\pm3.2}$ | $24.8_{\pm11.1}$ | $86.6_{\pm5.1}$ |
| | Yes | $48.3_{\pm17.2}$ | $23.3_{\pm3.7}$ | $95.7_{\pm2.7}$ | $13.6_{\pm2.6}$ | $34.2_{\pm14.3}$ | $84.0_{\pm10.9}$ |
| GraphSAGE-JK | No | $57.6_{\pm16.7}$ | $26.3_{\pm7.1}$ | $94.2_{\pm5.0}$ | $15.9_{\pm4.8}$ | $43.2_{\pm15.9}$ | $81.9_{\pm11.4}$ |
| | Yes | $\mathbf{58.3_{\pm8.3}}$ | $\mathbf{25.6_{\pm3.3}}$ | $\mathbf{95.8_{\pm1.5}}$ | $\mathbf{15.2_{\pm2.3}}$ | $\mathbf{42.5_{\pm8.9}}$ | $\mathbf{84.0_{\pm8.2}}$ |

