# OpenReview forum: "Tumor Budding T-cell Graphs: Assessing the Need for Resection in pT1 Colorectal Cancer Patients"
_MIDL.io/2023/Conference — MIDL 2023 Oral_

### Official Review · Reviewer_race · 2023-01-29

**Confidence:** 4
**Preliminary Rating:** 5
**Recommendation:** Oral, Poster

**Summary:**

In the paper, the authors introduce a new dataset for risk stratification of colorectal cancer patients. The dataset represents a collection of digital pathology slides with sparse labeling into two classes (tumor buds and immune system T-cells). In addition to the semantic labeling, the authors provide various graph representations reflecting the relationship between the tumor buds and T-cells. Finally, clinical meta-data such as lymphovascular invasion, tumor grade, etc. is provided.

The dataset is benchmarked using various graph-based neural nets to carry out risk stratification into low- and high-risk.

**Strengths:**

It is a very well-written paper. Was a pleasure to read! Thank you for your efforts!

The new digital pathology dataset (of 575 patients) is gonna be released upon acceptance that would contribute to the community working on graph-based medical imaging algorithms.

Proper benchmarking with existing SOTA methods for GNN is performed.

A detailed discussion of the experiments and the data is provided throughout the text and in the appendix.

**Weaknesses:**

My only concern is that such benchmarking of new datasets is typically published in journals (for example, since longer back-and-forth discussions with reviewers can provide more thorough feedback and reveal dataset details that could be useful to include in the paper). But if authors prefer to release it via MIDL, I will not object much.




**Deanonymize Review:**

no

**Paper Type:**

validation/application paper

**Questions To Address In The Rebuttal:**

"We create an ensemble of 10 models per experimental setup" How did the ten models differ?

I suggest authors improve figure 1 a bit as not everything was clear to me. I had to zoom in and find some common patterns across the 4 images to understand the relations thereof.
As I understood from my eye-based pattern matching, the bottom image in the middle (to which the "Tumor-budding hotspot selection" arrow leads) is a zoomed-in version of the 1st left-most image. The upper middle image (to which the "Tumor bud and T-cell detection" arrow leads)  is in turn a zoomed-in version of the bottom-middle one. If so, it would be nice to show graphically these relations similarly as you do in Fig. 2 (at least for the middle two images). Also, please mention what exactly are violet dots and light-green ones (i.e. tumor buds in green and T-cells in pink).

---

### Official Review · Reviewer_XSY8 · 2023-02-01

**Confidence:** 4
**Preliminary Rating:** 3
**Recommendation:** Poster

**Summary:**

The authors applied graph neural networks to extract spatial features from tumour budding and lymphocytic infiltration at the invasive front. The authors assessed various technical considerations in building a GNN classifier to predict high/low-risk patients. The GNN-based classifier is shown to potentially improve the risk assessment of patients, especially in reducing false positives in a mid-sized cohort.


**Strengths:**

- The paper has chosen a clinically relevant question, where significant improvement could lead to patient benefit.
- The authours have conducted a thorough ablation study, where a large number of technical variations were tested.

**Weaknesses:**

- While having a mid-size cohort in histology studies, the number of patients seems low considering training GNNs with very big graphs. In addition, the data is highly unbalanced. The risk of overfitting is high. In table 2 and the results in the appendix, the uncertainties in the key performance metric TNR are very high, suggesting widespread overfitting in the configurations tested.
- There is no data in this study suggesting the GNN approaches learned biologically meaningful signals that separate high vs low-risk patients. This again adds to the risk of the GNNs overfitting to circumstantial signals in the potentially noisy graphs.
- There are significant variations between the graph constructed by the three different approaches. The best model uses one that is drastically different from the other two. The implication of this is not clear.
- There is no non-GNN baseline to establish the added value of GNN approaches.

**Deanonymize Review:**

no

**Detailed Comments:**

While I appreciated the clinical value of the task, some epidemiological information on pT1 patients would help readers to better understand its significance.


**Paper Type:**

validation/application paper

**Questions To Address In The Rebuttal:**

- Providing evidence of generalisability. This could be accurate prediction in hold-out sets; improved prediction on difficult cases; learned biologically meaningful features of the spatial characteristics of tumour budding and infiltrating lymphocytes.
- What is the added value of GNN compared to other CNN/ViT-based approaches?

---

### Official Review · Reviewer_HMwf · 2023-02-05

**Confidence:** 4
**Preliminary Rating:** 4
**Recommendation:** Poster

**Summary:**

This paper investigates spatial layout of tumor buds and T-Cells as potential risk factors for colon cancer with deep graph models. Experiments were done with 626 tumors from 575 patients. Authors propose and compare three different graph structures. Specificity is increased by amount 20% compared to current convention.

**Strengths:**

-- exploring interaction between tumor buds and T-cells using GNN is innovative.
-- large data set
-- smooth and good structured paper
-- promising results with new directions
-- clinical problem is important



**Weaknesses:**

-- comparisons with SOTA not available
-- evaluation and model selection parts are not entirely clear
-- there is no external data available for generalization check
-- not clear how this can be embedded into routine clinics although this reviewer recognizes the importance of the work


**Deanonymize Review:**

no

**Detailed Comments:**

The paper is written good, overall innovation is solid as GNN was never applied to this problem before, and exploring new markers with DL sounds grounded. Results seem improved compared to baseline, but there is no comparison with any other method, and the evaluation routine about to chose best model is not clear.

**Paper Type:**

validation/application paper

**Questions To Address In The Rebuttal:**

-- is there other data available to blindly test the results for generalization understanding?
-- why there is no other deep network methods compared?
-- unclear notion on the choice of best model, it seems cherry picking for the model side
-- spatial exploration cannot be possible without GNN?

---

### Official Review · Reviewer_6aAk · 2023-02-05

**Confidence:** 4
**Preliminary Rating:** 4
**Recommendation:** Poster

**Summary:**

The authors propose to investigate two clinically relevant risk factors, namely tumor budding and lymphocyte infiltration at the invasive front. In order to capture the spatial layout of tumor buds and T-cells, the authors compare using three popular graph neural network architectures, as well as combinations of the node labels. Additionally, the others produced a new pT1 Hotspot Tumor Budding T-cell Graph (pT1-HBTG) dataset consisting of 626 tumor budding hotspots from 575 patients, which the authors have stated will be made open source.

**Strengths:**

1. The authors collected a new dataset which will be made open source.
2. The authors compare three popular GCN methods on this new dataset and demonstrate that the best of these is able to increase specificity by 20% compared to the currently recommended risk stratification based on histopathological risk factors, without losing any sensitivity.
3. The paper is well written, the analysis is thorough, and the reviewer is particularly impressed with the figures which were extremely helpful in demonstrating the utility of graph-based approaches for capturing these particular clinical features.

**Weaknesses:**

1. The choice to only report results for 10-ensemble models is a bit confusing. Ensembling is typically done to squeeze out some additional performance, but most papers do not report results only of ensembles. For practical clinical considerations, GCNs are already expensive to run compared to regular CNNs, and adding in a need to ensemble 10-models together makes this even more costly.

**Deanonymize Review:**

no

**Paper Type:**

validation/application paper

**Questions To Address In The Rebuttal:**

1. Why do the authors only report ensembled results with 10 models? It would be helpful to understand what gains come from the ensembling versus just a single GCN. I've never seen a paper ONLY report ensembled results, but rather show the baseline, then show the "best possible" model with the ensembling done.

---

### Official Review · Reviewer_NkWb · 2023-02-06

**Confidence:** 4
**Preliminary Rating:** 4
**Recommendation:** Poster

**Summary:**

The authors propose and compare three different graph structures, as well as different combinations of node labels, to capture the spatial layout of tumor buds and T-cells, in order to investigate them as potential risk predictors for CRC progression. The best-performing GNN architecture is able to increase specificity by 20% compared to the currently recommended risk stratification based on histopathological risk factors, without losing any sensitivity. The work was developed with a dataset of 626 tumor budding hotspots from 575 patients, that will be made available publicly.


**Strengths:**

- Overall good work
- Clear writing, with good English
- Extended ablation study and set of metrics for evaluation
- Interesting application of GNN, with promising results for risk assessment improvement

**Weaknesses:**

- Paper structure is not suitable to follow ideas easily
- Some key ideas are presented in Appendix
- The methodology is not well described in the main text
- A concrete proposal of the framework is missing



**Deanonymize Review:**

no

**Detailed Comments:**

- Section 2 should be in Appendix.
- Table 3 from Appendix B should be in section 3.
- Section 3: It would be better to have some summary on the workflow used for dataset creation, instead. Also, Figure 1 is not very informative (so, maybe, it can be removed?). Figure 2 should be referenced in Section 3.2
- Section 3.2.1: which kind of post-processing was used after Net?
- Section 3.2.1: how exactly was the output quality assessed?
- Section 3.2.1: typo in 200 x 200 pixels (letter x instead of times symbol)
- It is suggested to avoid to start paragraphs with Tables description. The authors can try to include reference to tables after some kind of explanation.
- Section 4.1: z is not defined.
- Figure 5 for Appendix C should be in the main text, as well as its description.
- All results tables are missing units.



**Paper Type:**

validation/application paper

**Questions To Address In The Rebuttal:**

- Authors should clarify the methodology.
- All F1 and TNR values are below 50%. Do the authors have any idea why the performance is so low?
- What is the final proposed methodology for risk assessment?

---

### Meta-Review · Area_Chair_JMje · 2023-02-23

**Recommendation:** Accept (Poster)
**Confidence:** 4

**Metareview:**

This paper proposed and compared different graph structures, as well as different combinations of node labels, to capture the spatial layout of tumor buds and T-cells, in order to investigate them as potential risk predictors for CRC progression.

Pros:
Interesting application with GCN with promising results;
Important clinical problem with clear writing;
Comparison with several methods over a large dataset;

Cons:
Some parts such as methods and experimental settings should be further elaborated;
Lack of non-GNN baselines for comparison;

In summary, we have most reviewers confirmed the merits of the paper on methodology contribution, clinical significance and detailed experimental comparison. Thus, a decision of accept is recommended.

---

### Meta-Review · Program_Chairs · 2023-02-28

**Recommendation:** Accept (Oral)
**Confidence:** 5

**Metareview:**

This an excellent and well-written study, with interest to the community. I recommend it for an oral presentation.